# Combined Evaluation of Geriatric Nutritional Risk Index and Modified Creatinine Index for Predicting Mortality in Patients on Hemodialysis

**DOI:** 10.3390/nu14040752

**Published:** 2022-02-10

**Authors:** Takahiro Yajima, Kumiko Yajima, Maiko Arao

**Affiliations:** 1Department of Nephrology, Matsunami General Hospital, Gifu 501-6062, Japan; matsunamijinzou@gmail.com; 2Department of Internal Medicine, Matsunami General Hospital, Gifu 501-6062, Japan; green_tea_1324@yahoo.co.jp

**Keywords:** geriatric nutritional risk index (GNRI), modified creatinine index (mCI), hemodialysis, protein-energy wasting (PEW), all-cause mortality

## Abstract

The geriatric nutritional risk index (GNRI) and modified creatinine index (mCI) are surrogate markers of protein-energy wasting in patients receiving hemodialysis. We aimed to examine whether a combined evaluation of these indices improved mortality prediction in this population. We retrospectively investigated 263 hemodialysis patients divided into two groups, using 91.2 and 20.16 mg/kg/day as cut-off values of GNRI and mCI, respectively. The resultant four groups were reshuffled into four subgroups defined using combinations of cut-off values of both indices and were followed up. During the follow-up period (median: 3.1 years), 103 patients died (46/103, cardiovascular causes). Lower GNRI and lower mCI were independently associated with all-cause mortality (adjusted hazard ratio (aHR) 4.96, 95% confidence intervals (CI) 3.10–7.94, and aHR 1.92, 95% CI 1.22–3.02, respectively). The aHR value for the lower GNRI and lower mCI group vs. the higher GNRI and higher mCI group was 7.95 (95% CI 4.38–14.43). Further, the addition of GNRI and mCI to the baseline risk assessment model significantly improved the C-index of all-cause mortality (0.801 to 0.835, *p* = 0.025). The simultaneous evaluation of GNRI and mCI could be clinically useful to stratify the risk of mortality and to improve the predictability of mortality in patients on hemodialysis.

## 1. Introduction

Protein-energy wasting (PEW) is a common malnutritional form that is characterized by a loss of fat and muscle mass caused by chronic inflammation and is related to an increased risk of all-cause mortality in patients receiving hemodialysis (HD) [1]. Several assessment methods of nutrition have been employed for the evaluation of PEW in this population. The malnutrition inflammation score (MIS) is one of the most validated tools for evaluating patients on HD at a risk of malnutrition [2,3,4]. However, assessments by well-trained examiners are needed to obtain a consistent MIS [5]. Conversely, objective screening methods such as the geriatric nutritional risk index (GNRI) and the modified creatinine index (mCI) are also frequently used for evaluating nutritional statuses in this vulnerable patient population [6,7,8]. While GNRI estimation is performed using the serum albumin level (Alb) and the body mass index (BMI) [6,9], mCI is calculated using parameters including age, sex, urea clearance (Kt/V), and pre-hemodialysis serum creatinine level [7]. These objective indices are easily calculable using biochemical values obtained from routine blood tests. A meta-analysis concluded that GNRI is a clinically useful predictor of all-cause and cardiovascular mortality in HD patients [10]. Some studies have reported that mCI also correlates with both increased risks of cardiovascular events and with all-cause and cardiovascular mortality [11,12,13]. Moreover, Yamada et al. recently demonstrated that the predictive accuracy of GNRI for all-cause mortality was equal to that of mCI in patients on HD [14]. Furthermore, we recently demonstrated that a combination of GNRI and the erythropoietin resistance index or the extracellular/intracellular fluid ratio measured using bioimpedance analysis could improve the predictability of mortality risk in patients on HD [15,16].

Therefore, we hypothesized that a combination of GNRI and mCI scores could be utilized to improve the predictive accuracy of mortality in patients receiving HD. The present study aimed to investigate whether a combined estimation of GNRI and mCI indices could improve the predictability of all-cause mortality risks in patients on HD.

## 2. Materials and Methods

### 2.1. Study Population

This retrospective cohort study enrolled 263 patients who underwent HD for ≥6 months (with no hospitalization for at least the last 3 months) at the outpatient clinic of the Matsunami General Hospital (Kasamatsu, Gifu, Japan) between April 2008 and March 2021. Data were collated via retrospective chart review.

This study adhered to the guidelines of the Declaration of Helsinki and was approved by the ethics committee of the Matsunami General Hospital (Approval no. 504). The requirements of informed consent were waived because all patients’ data were anonymized before access.

### 2.2. Collection of Data

Data of all study participants were obtained from their medical charts: age; sex; underlying renal disease (diabetic nephropathy, chronic glomerulonephritis, nephrosclerosis, and others); type of vascular access (arteriovenous fistula/graft or central venous catheter), HD vintage; alcohol and/or smoking habit; history of comorbidities (hypertension, diabetes mellitus, cardiovascular disease (atherosclerotic/ischemic heart disease, peripheral vascular disease, transient ischemic attack/cerebrovascular accident, congestive heart failure, other cardiac diseases endocarditis, myocarditis, pericarditis, cardiac devices, valve replacement, and heart transplantation, and dysrhythmia), lung disease (chronic obstructive pulmonary disease), liver disease, gastrointestinal bleeding, and cancer); height; dry weight; and use of RAS inhibitors or statins. In the present study, patients receiving antihypertensive medications, those with a diastolic blood pressure ≥90 mmHg, and those with a systolic blood pressure ≥140 mmHg (measured before a HD session) were identified as having hypertension. Diabetes mellitus was recorded in those with a history of the disease and/or in those using glucose-lowering medications. The dialysis patient-specific comorbidity index including 11 comorbid conditions, which Liu et al. proposed, was scored as previously reported [17,18]. The assigned scoring weights were as follows: 1 point for diabetes and atherosclerotic/ischemic heart disease; 3 points for congestive heart failure; and 2 points for the remaining comorbidities. Blood tests were performed before the first HD session of the week, on either Monday or Tuesday.

### 2.3. Calculation of Nutritional Indices and Patient Grouping

The GNRI was calculated using the following formula [6,9]:

GNRI = 14.89 × Alb (g/dL) + 41.7 × Dry weight (kg)/ideal body weight (kg) (=height^2^ [m^2^] × 22) = 14.89 × Alb (g/dL) + 41.7 × BMI/22.

When BMI was ≥22 kg/m^2^, the value of BMI/22 was considered as 1.

The mCI was calculated using parameters including sex, age, spKt/V (for urea clearance), and pre-hemodialysis creatinine level using the following formula:

mCI (mg/kg/day) = 16.21 + 1.12 × (0 for woman; 1 for man) − 0.06 × age (years) − 0.08 × spKt/V for urea + 0.009 × pre-hemodialysis creatinine (μmol/L) [7].

Patients were divided into 2 groups (high vs. low GNRI groups) using a cut-off GNRI value of 91.2, as reported previously [6]. They were also divided into 2 groups (high vs. low mCI groups) using the cut-off median mCI value of 20.16 mg/kg/day. Finally, the resultant 4 patient groups were redistributed into 4 subgroups according to a combination of the cut-off values of those indices: Group 1 (G1), high GNRI and high mCI; Group 2 (G2), high GNRI and low mCI; Group 3 (G3), low GNRI and high mCI; Group 4 (G4), low GNRI and low mCI.

### 2.4. Study Endpoint and Patient Follow-Up

The study endpoint was all-cause mortality. Patients were included in the study from April 2008 to March 2020 and were followed up until March 2021. As per protocol, the follow-up period was defined as the time from the date of the blood tests (used for calculating the indices) to the date of either death or transfer to a renal transplantation or another HD facility. Surviving patients were censored in March 2021.

### 2.5. Statistical Analysis

Non-normally and normally distributed variables were expressed as medians (interquartile range) and as means ± standard deviations, respectively. The differences among the four patient subgroups derived using a combination of cut-off GNRI and mCI values were compared using the Kruskal–Wallis test or a one-way analysis of variance for continuous variables and the Chi-squared test for categorical variables. The Kaplan–Meier method was used to evaluate the survival rates, and the log-rank test was used to compare the inter-group differences. Cox regression analysis was performed to calculate the hazard ratio (HR) and the 95% confidence intervals (CIs) for all-cause mortality. The multivariate Cox regression model was applied to variables reported to be established risk indicators for mortality in patients receiving HD. In this case, the type of vascular access, HD vintage, concurrent hypertension, use of RAS inhibitors or statins, comorbidity index, hemoglobin level, and log C-reactive protein (CRP) were included as covariates. Because serum Alb and BMI values were required for GNRI estimation and the patient’s sex, age, urea clearance, and serum creatinine level were required for mCI calculation, these variables were excluded from the covariate analysis.

Regarding model discrimination, the C-index, which was defined as the area under the receiver operating characteristic curve in a logistic regression model, was calculated to investigate the predictive accuracy of all-cause mortality [19]. In this study, the C-indices were compared between a baseline risk model including the established classical risk factors and models of adding the GNRI alone, the mCI alone, and a combination of GNRI and mCI. Furthermore, the net reclassification improvement (NRI) and the integrated discrimination improvement (IDI) were also calculated. The NRI is defined as the relative improvement of the number of patients for predicting mortality, whereas the IDI is defined as the average improvement of predicting mortality [20]. We used the IBM SPSS version 24 software (IBM Corp., Armonk, NY, USA) to perform the statistical analyses. A *p*-value less than 0.05 denoted statistical significance.

## 3. Results

### 3.1. Baseline Characteristics of the Study Participants

Data on the baseline characteristics of the participants are shown in Table 1. The mean age of the study population was 63.8 ± 13.7 years, and 66.5% of the participants were men. The types of vascular access comprised arteriovenous fistulas (201; 76.4%), arteriovenous grafts (58; 22.1%), and central venous catheters (4; 1.5%). The dialysis vintage value was 1.5 (0.7–4.3) years, with 94.3% and 47.1% patients having hypertension and diabetes, respectively. The comorbidity index was 4.4 ± 3.6. Furthermore, 57.8% and 44.1% of the patients were taking RAS inhibitors and statins, respectively. The mean BMI was 22.1 ± 4.2 kg/m^2^. The hemoglobin level, serum Alb, CRP, and spKt/V for urea were 10.6 ± 1.4 g/dL, 3.6 ± 0.4 g/dL, 0.15 (0.06–0.43) mg/dL, and 1.34 ± 0.29, respectively. The GNRI and mCI values were 93.1 ± 7.6 and 20.2 ± 3.0 (20.16 (17.87–22.18)) mg/kg/day, respectively.

### 3.2. Associations of GNRI and mCI with All-Cause Mortality

A total of 103 deaths were noted during a median follow-up period 3.1 years (1.5–6.0); these included 46 (44.7%), 30 (29.1%), 13 (12.6%), and 14 (13.6%) deaths due to cardiovascular disease (19: sudden cardiac death or fatal arrhythmia, 15: congestive heart failure, 9: cerebrovascular accident, and 3: myocardial infarction), infection, malignancies, and other causes, respectively (Figure 1).

The multivariate Cox regression analysis adjusted by HD vintage, hypertension, the type of vascular access, use of RAS inhibitors and statins, the comorbidity index, hemoglobin level, and log CRP revealed that the GNRI and mCI were significant indictors for all-cause mortality (GNRI: adjusted HR (aHR), 0.89; 95% CI, 0.86–0.92, *p* < 0.0001; mCI aHR, 0.83; 95% CI, 0.76–0.90, *p* < 0.0001) (Table 2). Among patients divided into two groups using the GNRI cut-off value of 91.2, the 7-year survival rates were 24.8% and 68.4% in those with lower and higher indices, respectively (*p* < 0.0001) (Figure 2a). Similarly, among patients divided into two groups using the mCI cut-off value of 20.16 mg/kg/day, the 7-year survival rates were 39.9% and 67.7% in those with lower and higher values, respectively (*p* < 0.0001) (Figure 2b). The aHR for all-cause mortality were 4.96 (95% CI, 3.10–7.94, *p* < 0.0001) and 1.92 (95% CI, 1.22–3.02, *p* = 0.0047) for low GNRI vs. high GNRI and for low mCI vs. high mCI, respectively (Table 2). With the combined application of both indices, 7-year survival rates were 74.4% in G1, 57.6% in G2, 38.7% in G3, and 20.9% in G4 (*p* < 0.0001) (Figure 2c). The aHR for G4 vs. G1 was 7.95 (95% CI, 4.38–14.43, *p* < 0.0001) (Table 3).

### 3.3. Model Discrimination in Predicting All-Cause Mortality

When comparing with a traditional risk model accounting for classical risk indictors (HD vintage, hypertension, type of vascular access, use of RAS inhibitors and statins, the comorbidity index, hemoglobin level, and log CRP), adding the GNRI and mCI assessment significantly improved the C-index from 0.801 to 0.835 (*p* = 0.025) (Table 3). The NRI and IDI also significantly improved upon the inclusion of GNRI and mCI indices in the evaluation (0.491 (*p* = 0.00006) and 0.058 (*p* = 0.0001), respectively) (Table 3).

## 4. Discussion

Lower values of GNRI and mCI were, as previously reported, independently associated with an increased risk of all-cause mortality, respectively. Further, in the present study, patients on HD in the group of combined lower GNRI and mCI values were found to be at the highest risk of all-cause mortality. Moreover, the C-index of all-cause mortality significantly improved when the combination of the GNRI and mCI was added to the baseline model with classical risk indicators. These findings suggested that the combined utilization of both indices could be clinically useful to improve the predictability of all-cause mortality as well as to stratify the risks of all-cause mortality in patients on HD. Therefore, we recommend evaluating both indices at the same time in patients receiving HD.

PEW, a malnutritional state defined by a loss of fat and muscle mass because of catabolic inflammation, is prevalent in patients receiving HD and is also related to increased risks of mortality [4,21,22,23,24]. The MIS proposed by Kalanter-Zadeh et al. is the gold standard method for assessing the nutritional status of HD patients [2]. Although MIS assessment is a useful predictor of morbidity and mortality in this population, its application necessitates subjective evaluations by well-trained examiners [5]. Conversely, the GNRI, which is easily calculable with serum Alb level and BMI [6,8], is an objective tool for the longitudinal assessment of the nutritional status [25]. Yamada et al. reported that there was a negative and significant correlation between the GNRI and MIS [6]. They reported 91.2 as the cut-off value of GNRI, which corresponded to malnutrition associated with an MIS > 6 [6]. This cut-off GNRI value was used in the present study. A meta-analysis recently performed by Xiong et al. demonstrated that the GNRI is a clinically useful indictor of predicting all-cause and cardiovascular mortality in patients on HD [10]. Some previous studies showed that the predictive accuracy of MIS was better than that of GNRI [4,5]. However, Chen et al. recently conducted a large cohort study and reported that the predictability of all-cause and cardiovascular mortality of GNRI was comparable to that of MIS [26]. The mCI is an indicator of muscle wastage and an objective nutritional assessment tool used in HD patients [7,27,28]. Hwang et al. reported that the mCI was correlated with the GNRI [12], and Tsai et al. recently found that the mCI is an independent predictor of PEW [29]. Furthermore, some studies reported that the mCI is a useful tool to predict the occurrence of new cardiovascular events and that of infection-associated or all-cause mortality [11,12,13,30]. Therefore, both the GNRI and mCI are established indicators of PEW and are clinically useful, independent predictors of mortality in patients undergoing HD.

We recently demonstrated that a combined assessment of bio-impedance analysis-estimated extracellular fluid/intracellular fluid ratio and GNRI could improve the predictive accuracy for mortality in patients on HD [15]. We also reported that both the erythropoietin resistance index and GNRI were independently correlated with all-cause mortality, and the combined evaluation of these indices improved the predictability of mortality risk in this population [16]. Recently, Yamada et al. showed that the GNRI and mCI were equally useful tools for predicting all-cause mortality in a patient population receiving HD [14]. We therefore hypothesized that a combination of these indices could improve the predictability of mortality.

In this study, patients with low GNRI and low mCI were independently associated with increased risks of all-cause mortality, respectively. Furthermore, patients with lower values of both GNRI and mCI (G4) demonstrated the highest risk of mortality. Moreover, adding both indices to the baseline evaluation model consisting of classical risk factors led to the significantly improved predictably of all-cause mortality, as observed in the discrimination analysis model. Thus, the combination of GNRI and mCI may be useful to stratify risks of all-cause mortality and to allow clinicians to predict the mortality risk more precisely in patients on HD. The GNRI and mCI employ different evaluation parameters and may be considered surrogate indicators of PEW. Therefore, the simultaneous evaluation of both indices may be recommended as a more robust and accurate clinical assessment tool.

This study has some limitations. First, this is a retrospective, single-center study that included a relatively small number of patients on HD. Because this study is observational, it cannot provide direct cause-and-effect risk associations. Second, the present study only focused on the Japanese population, who are known to have better prognoses compared to those of their counterparts in both the USA and Europe [31]. Therefore, our study results might not be applicable to HD patients in other countries. Third, only the associations of the baseline values of the GNRI and mCI with all-cause mortality were evaluated; therefore, any changes of these indices were not considered during this study’s follow-up period. Moreover, during the follow-up period, 28 patients were treated with an enteral diet, and four patients were temporally treated with intradialytic parenteral nutrition at the discretion of the attending physician. These nutritional therapies might have affected the results of the present study. Fourth, we could not include data on the residual kidney function. The mCI is developed on the assumption that patients have no residual kidney function; therefore, residual kidney function may affect the mCI value. To validate our findings, further large-scale, prospective, multicenter studies may be required.

## 5. Conclusions

Low values of GNRI and mCI were independently associated with increased risks of all-cause mortality, and patients on HD with combined low values of both indices demonstrated the highest all-cause mortality risk. The predictability of all-cause mortality improved by adding the combined GNRI and mCI evaluation to the baseline assessment model that included established risk factors. The combined assessment of GNRI and mCI may be clinically helpful to stratify and precisely predict the all-cause mortality risk in patients on HD. Therefore, we propose the simultaneous evaluation of both these indices in patients receiving HD.

## Figures and Tables

**Figure 1 nutrients-14-00752-f001:**
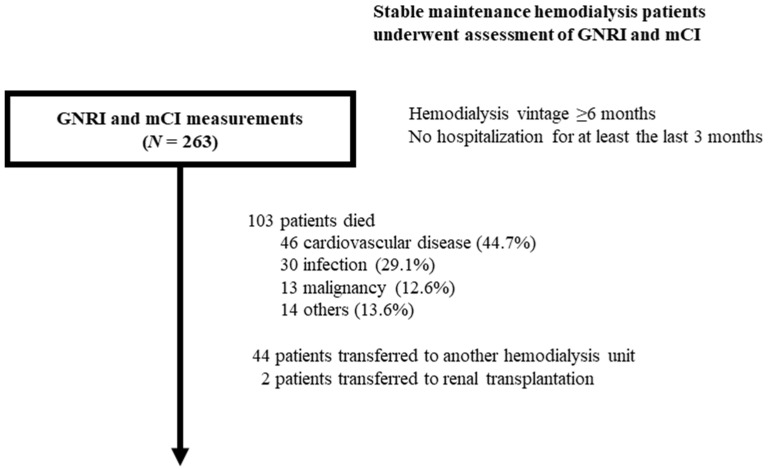
Flow diagram of the present study.

**Figure 2 nutrients-14-00752-f002:**
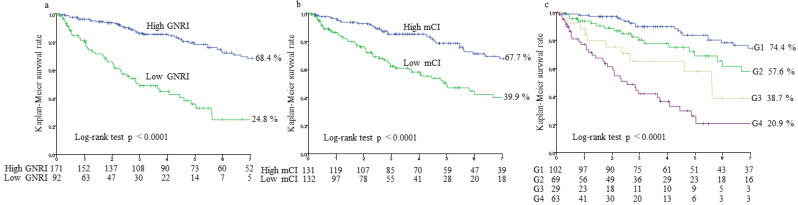
Kaplan–Meier survival curves for all-cause mortality. All-cause mortality rates for two groups of geriatric nutritional risk index (GNRI) < 91.2 vs. GNRI ≥ 91.2 (**a**), two groups of modified creatinine index (mCI) < 20.16 mg/kg/day vs. mCI ≥ 20.16 mg/kg/day (**b**), and four groups of combined GNRI and mCI (**c**). G1 (group 1), GNRI ≥ 91.2 and mCI ≥ 20.16 mg/kg/day; G2 (group 2), GNRI ≥ 91.2 and mCI < 20.16 mg/kg/day; G3 (group 3), GNRI < 91.2 and mCI ≥ 20.16 mg/kg/day; and G4 (group 4), GNRI < 91.2 and mCI < 20.16 mg/kg/day.

**Table 1 nutrients-14-00752-t001:** Baseline patient characteristics.

	All Patients (*N* = 263)	G1 (*N* = 102)	G2 (*N* = 69)	G3 (*N*= 29)	G4 (*N* = 63)	*p*-Value
Age (years)	63.8 ± 13.7	56.0 ± 14.2	66.8 ± 9.1	62.4 ± 12.9	73.8 ± 9.5	<0.0001
Men (%)	66.5	76.5	59.4	82.8	50.8	0.0009
Underlying renal disease						0.012
Diabetic nephropathy (%)	42.6	42.7	58.9	9.1	38.8	
Chronic glomerulonephritis (%)	29.6	31.3	14.3	54.5	32.7	
Nephrosclerosis (%)	20.2	19.8	16.1	27.2	22.4	
Others (%)	7.4	6.3	10.7	9.1	6.1	
Type of vascular access						0.0097
Arteriovenous fistula (%)	76.4	80.4	71.0	89.7	69.8	
Arteriovenous graft (%)	22.1	19.6	29.0	10.3	23.8	
Central venous catheter (%)	1.5	0	0	0	6.3	
Hemodialysis vintage (years)	1.5 (0.7–4.3)	3.7 (1.2–6.2)	1.0 (0.7–2.0)	2.5 (0.8–5.5)	0.8 (0.6–1.7)	<0.0001
Alcohol (%)	25.8	25.5	29.0	34.4	19.0	0.39
Smoking (%)	27.8	28.4	37.7	37.9	11.1	0.0035
Hypertension (%)	94.3	95.1	97.1	96.6	88.9	0.18
Comorbidity index	4.4 ± 3.6	3.5 ± 3.5	5.6 ± 3.5	3.6 ± 3.6	4.9 ± 3.3	0.0007
Diabetes mellitus (%)	47.1	42.2	65.2	27.6	44.4	0.0021
Atherosclerotic/ischemic heart disease (%)	29.7	24.5	44.9	20.7	25.4	0.017
Peripheral vascular disease (%)	14.1	10.8	20.3	10.3	14.3	0.34
Transient ischemic attack/cerebrovascular accident (%)	16.7	16.7	15.9	17.2	17.5	0.99
Congestive heart failure (%)	57.4	44.1	72.5	48.3	66.7	0.0006
Other cardiac disease (%)	23.2	17.6	29.0	13.8	30.2	0.096
Dysrhythmia (%)	11.4	15.7	14.5	3.4	4.8	0.043
Chronic obstructive pulmonary disease (%)	3.4	2.0	5.8	3.4	3.2	0.62
Liver disease (%)	6.1	2.9	5.8	13.8	7.9	0.19
Gastrointestinal bleeding (%)	6.5	3.9	7.2	10.3	7.9	0.54
Cancer (%)	13.3	6.9	15.9	6.9	23.8	0.012
RAS inhibitor usage (%)	57.8	56.9	62.3	62.1	52.4	0.66
Statin usage (%)	44.1	46.0	53.6	31.0	36.5	0.10
Height (cm)	161 ± 9	163 ± 8	159 ± 8	164 ± 8	157 ± 9	<0.0001
Dry weight (kg)	57.6 ± 13.4	62.1 ± 13.1	61.4 ± 12.2	54.2 ± 12.6	47.7 ± 9.2	<0.0001
Body mass index (kg/m^2^)	22.1 ± 4.2	23.1 ± 3.8	24.1 ± 3.9	19.9 ± 3.3	19.4 ± 3.5	<0.0001
Blood urea nitrogen (mg/dL)	56.7 ± 14.2	62.7 ± 12.8	52.2 ± 11.8	57.5 ± 12.7	51.5 ± 15.7	<0.0001
Creatinine (mg/dL)	9.0 ± 3.1	11.6 ± 2.2	6.8 ± 1.8	10.7 ± 1.3	6.5 ± 1.7	<0.0001
Alb (g/dL)	3.6 ± 0.4	3.9 ± 0.3	3.8 ± 0.3	3.3 ± 0.3	3.3 ± 0.4	<0.0001
Hb (g/dL)	10.6 ± 1.4	10.8 ± 1.4	10.5 ± 1.2	10.4 ± 1.4	10.3 ± 1.5	0.23
Total cholesterol (mg/dL)	148 ± 33	149 ± 33	159 ± 35	135 ± 30	141 ± 30	0.0013
Uric acid (mg/dL)	6.9 ± 1.7	7.4 ± 1.8	6.7 ± 1.5	6.7 ± 1.7	6.4 ± 1.8	0.0051
Ca (mg/dL)	8.9 ± 0.8	9.2 ± 0.7	8.7 ± 0.6	8.6 ± 0.8	8.7 ± 1.0	<0.0001
P (mg/dL)	4.9 ± 1.4	5.4 ± 1.5	4.5 ± 1.0	5.3 ± 1.0	4.4 ± 1.3	<0.0001
Intact parathyroid hormone (pg/mL)	115 (51–183)	116 (50–186)	126 (67–185)	115 (58–211)	100 (29–169)	0.59
Glucose (mg/dL)	144 ± 62	144 ± 66	160 ± 71	130 ± 44	132 ± 47	0.046
CRP (mg/dL)	0.15 (0.06–0.43)	0.13 (0.06–0.28)	0.12 (0.06–0.35)	0.54 (0.11–1.12)	0.21 (0.07–1.30)	0.0050
spKt/V for urea	1.34 ± 0.29	1.32 ± 0.28	1.27 ± 0.28	1.43 ± 0.28	1.40 ± 0.31	0.018
GNRI	93.1 ± 7.6	98.1 ± 3.9	96.7 ± 3.5	86.0 ± 4.7	84.4 ± 6.1	<0.0001
mCI	20.2 ± 3.0	22.8 ± 2.2	18.1 ± 1.4	21.6 ± 1.3	17.4 ± 1.7	<0.0001

(Abbreviations: Alb: albumin, Hb: hemoglobin, Ca: calcium, P: phosphorus, CRP: C-reactive protein, GNRI: geriatric nutritional risk index, mCI: modified creatinine index). G1 (group 1), GNRI ≥ 91.2 and mCI ≥ 20.16 mg/kg/day; G2 (group 2), GNRI ≥ 91.2 and mCI < 20.16 mg/kg/day; G3 (group 3), GNRI < 91.2 and mCI ≥ 20.16 mg/kg/day; and G4 (group 4), GNRI < 91.2 and mCI < 20.16 mg/kg/day.

**Table 2 nutrients-14-00752-t002:** Cox analysis of GNRI and mCI for all-cause mortality.

	Univariate Analysis		Multivariate Analysis *	
Variables	HR (95% CI)	*p*-Value	HR (95% CI)	*p*-Value
GNRI (continuous)	0.89 (0.87–0.91)	<0.0001	0.89 (0.86–0.92)	<0.0001
mCI (continuous)	0.81 (0.75–0.87)	<0.0001	0.83 (0.76–0.90)	<0.0001
Lower GNRI	4.26 (2.82–6.43)	<0.0001	4.96 (3.10–7.94)	<0.0001
Lower mCI	2.51 (1.68–3.74)	<0.0001	1.92 (1.22–3.02)	0.0047
Cross-classified (vs. G1)		<0.0001		<0.0001
G2	1.99 (1.16–3.42)	0.013	1.11 (0.60–2.03)	0.75
G3	3.71 (1.89–7.27)	0.0001	2.75 (1.31–5.76)	0.0073
G4	7.23 (4.25–12.32)	<0.0001	7.95 (4.38–14.43)	<0.0001

Abbreviations: GNRI: geriatric nutritional risk index, mCI: modified creatinine index. * adjusted by HD vintage, hypertension, the type of vascular access, use of RAS inhibitors and statins, the comorbidity index, hemoglobin level, and log CRP.

**Table 3 nutrients-14-00752-t003:** Predictive accuracies of GNRI and mCI for all-cause mortality.

Variables	C-Index	*p*-Value	NRI	*p*-Value	IDI	*p*-Value
Classical risk factors *	0.801 (0.748–0.855)		Ref.			Ref.
+ GNRI	0.828 (0.777–0.878)	0.061	0.399	0.0012	0.048	0.00051
+ mCI	0.822 (0.771–0.872)	0.083	0.391	0.0016	0.034	0.0028
+ GNRI and mCI	0.835 (0.786–0.884)	0.025	0.491	0.00006	0.058	0.0001

Abbreviations: GNRI: geriatric nutritional risk index, mCI: modified creatinine index, Ref.: reference values. * containing HD vintage, hypertension, the type of vascular access, use of RAS inhibitors and statins, the comorbidity index, hemoglobin level, and log CRP.

## Data Availability

The data used for analysis in the present study are available upon request to the corresponding author. The data are not publicly available because of privacy or ethical restrictions.

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
