# Peer review of "Combined Evaluation of Geriatric Nutritional Risk Index and Modified Creatinine Index for Predicting Mortality in Patients on Hemodialysis"

_nutrients, 2022, doi:10.3390/nu14040752_

Round 1

Reviewer 1 Report

The paper is well written and the analysis is clear. I have some additional comments to the authors:

  • Please provide a flow diagram of patient enrollment. How many patients were screened, the reasons why patients were excluded, how many patients were transplanted, how many were switched to PD, etch.
  • The all-cause death is a clear outcome. In contrast, the cardiovascular death requires adjudication of the event. However, this study followed a retrospective design, and the etiology of mortal events was not prospectively adjudicated.
  • The authors do not report specific inclusion/exclusion criteria, with the exception of dialysis vintage. This issue requires clarification. For example, patients with a recent hospitalization before the baseline evaluation, patients with a recent CV event, patients with infectious complications at baseline were included or not?
  • Statistical analysis: the multivariate models included some prespecified covariates. However, there is substantial residual confounding in this analysis. Several parameters that affect the mortality were not considered: vascular access, presence of substantial residual kidney function at baseline, use of cardioprotective therapies (RAS-blockers, statins, etch). The authors should address this important issue in their analysis.
  • How many patients were being treated with nutritional supplements during dialysis or outside of dialysis?
  • Limitations: this study is observational and can not provide direct cause-and-effect risk associations – this limitation needs to be acknowledged.

Author Response

Response to the Reviewer #1

Thank you very much for your constructive comments.

Please provide a flow diagram of patient enrollment. How many patients were screened, the reasons why patients were excluded, how many patients were transplanted, how many were switched to PD, etch.

Thank you very much for the comments.

At our outpatient clinic, only stable patients on hemodialysis were followed up. Therefore, there are no patients who underwent kidney transplantation or switched to peritoneal dialysis. In this retrospective study, we included all patients who underwent stable hemodialysis for ≥6 months with no hospitalization for at least the last 3 months. During the follow-up period, 44 patients were transferred to another hemodialysis unit and 2 patients were transferred for a kidney transplantation. We have added a flow diagram, for an improved understanding of this.

The all-cause death is a clear outcome. In contrast, the cardiovascular death requires adjudication of the event. However, this study followed a retrospective design, and the etiology of mortal events was not prospectively adjudicated.

Thank you very much for the comments.

We completely understand your concerns and agree with you. We believe that due to the retrospective study design, cardiovascular deaths may be more difficult to determine with absolute clarity. Therefore, we have deleted the data on cardiovascular mortality form our manuscript.

The authors do not report specific inclusion/exclusion criteria, with the exception of dialysis vintage. This issue requires clarification. For example, patients with a recent hospitalization before the baseline evaluation, patients with a recent CV event, patients with infectious complications at baseline were included or not?

Thank you for these comments.

We understand that it is important to clarify the inclusion/exclusion criteria of this study. As we have mentioned in a response above, in the present retrospective study, we included all patients who underwent stable hemodialysis at the outpatient clinic for ≥6 months with no hospitalization for at least the last 3 months. We have now mentioned this in the revised manuscript.

Statistical analysis: the multivariate models included some prespecified covariates. However, there is substantial residual confounding in this analysis. Several parameters that affect the mortality were not considered: vascular access, presence of substantial residual kidney function at baseline, use of cardioprotective therapies (RAS-blockers, statins, etch). The authors should address this important issue in their analysis.

Thank you for your insightful comment.

We agree with your suggestions. We added details on the type of vascular access and use of RAS-inhibitors and statins to Table 1. The same factors were included in the multivariate models. Furthermore, in accordance with the suggestion from Reviewer #2, we have also included the Charlson's comorbidity index in the multivariate models. Because this index includes diabetes and cardiovascular-specific comorbidity (ischemic heart disease, congestive heart failure, peripheral vascular disease, cerebrovascular disease, dysrhythmia, and other cardiac diseases), we excluded the history of diabetes and cardiovascular disease from the models. Due to the retrospective nature of the study, we could not include the baseline residual kidney function. However, we believe that this is an important factor that may also affect the mCI, therefore we have addressed this as a limitation of the study in the Discussion section.

How many patients were being treated with nutritional supplements during dialysis or outside of dialysis?

Thank you for this comment.   

At the discretion of the attending physician, during the follow-up period, 28 patients were put on an enteral diet and four patients temporally received intradialytic parenteral nutrition. These nutritional therapies might have affected the results of the present study; therefore, this point has been addressed as a limitation of the study.

Limitations: this study is observational and can not provide direct cause-and-effect risk associations – this limitation needs to be acknowledged.

Thank you for flagging this with us.

We believe that this is a very important point; accordingly, we have addressed this as a limitation of the study.

Reviewer 2 Report

In this well and clearly written article, Japanese nephrologists of a single center analyzed retrospectively their cohort of 263 hemodialysis patients followed for 3 years and showed that a lower Geriatric Nutritional Risk Index and a lower Modified Creatinine Index were independently predictive of all cause mortality and also cardiovascular death. Additionally, to the well-known confounding factors analyzed by the authors in the multivariate Cox proportional hazards, analyses should also be adjusted on age because age of G4 group is far higher (73.8 years) and also  the Charlson's comorbidity index shown to be highly predictive of death in both hemodialysis and peritoneal dialysis patients.

Author Response

Response to the Reviewer #2

Thank you very much for your constructive comments.

In this well and clearly written article, Japanese nephrologists of a single center analyzed retrospectively their cohort of 263 hemodialysis patients followed for 3 years and showed that a lower Geriatric Nutritional Risk Index and a lower Modified Creatinine Index were independently predictive of all cause mortality and also cardiovascular death. Additionally, to the well-known confounding factors analyzed by the authors in the multivariate Cox proportional hazards, analyses should also be adjusted on age because age of G4 group is far higher (73.8 years) and also the Charlson's comorbidity index shown to be highly predictive of death in both hemodialysis and peritoneal dialysis patients.

Response:

Thank you very much for your comments.

We agree with your assessment. Accordingly, we included the comorbidity index which was modified for assessing patients on dialysis (Liu et al., Kidney Int. 2010 Jan;77(2):141-51.) in the multivariate Cox proportional hazard model. Because the comorbidity index includes diabetes and cardiovascular diseases, we excluded these variables from the multivariate Cox proportional hazard model.

As you have rightly mentioned, the age in the G4 group (low GNRI and low mCI) was high. However, as we have mentioned in the methods section, the mCI itself is a powerful predictor that already accounts for the age, as follows: mCI (mg/kg/day) = 16.21 + 1.12 × (0 for woman; 1 for man) – 0.06 × age (years) – 0.08 × single pool Kt/V for urea + 0.009 × pre-hemodialysis serum creatinine (μmol/L). Thus, the mCI was significantly negatively correlated with the age (r = -0.71, p <0.0001). Moreover, when the mCI was doubly adjusted for age, its significance for predicting all-cause mortality was lost (adjusted HR 0.92 95%CI 0.83-1.01, p=0.080). We believe that this adjustment may be statistically inappropriate in this study, therefore we did not include age in the multivariate Cox proportional hazard model. Thank you for your kind understanding this.

Round 2

Reviewer 1 Report

The authors have performed additional analyses and have clearly acknowledged the limitations of their study - i have no further comments.

Reviewer 2 Report

This revised version has taken into account as possible reviewers' comments.